# Ultrasound-Guided Radiofrequency Ablation for Primary Hyperparathyroidism Induced by Multiple Endocrine Neoplasia 1—A Case Report

**DOI:** 10.3390/diagnostics12102553

**Published:** 2022-10-20

**Authors:** Zhenping Han, Tingting Li, Siyi Wang, Li Gao, Ying Hu, Yu Zhao, Jieping Yan

**Affiliations:** 1Center for Clinical Pharmacy, Department of Pharmacy, Zhejiang Provincial People’s Hospital, Affiliated People’s Hospital, Hangzhou Medical College, Hangzhou 310014, China; 2Department of Pharmacology, College of Pharmaceutical Sciences, Zhejiang University of Technology, Hangzhou 310014, China; 3Key Laboratory of Endocrine Gland Diseases of Zhejiang Province, Hangzhou 310014, China; 4Department of Endocrinology, Zhejiang Provincial People’s Hospital, Affiliated People’s Hospital, Hangzhou Medical College, Hangzhou 310014, China

**Keywords:** multiple endocrine neoplasia type 1, primary hyperparathyroidism, radiofrequency ablation, ultrasound-guided, parathyroid

## Abstract

Multiple endocrine neoplasia type 1 (MEN1) is a syndrome characterized by the occurrence of two or more endocrine gland tumors. Here, we show a case of a 52-year-old man diagnosed with MEN1 through gastrinoma, parathyroid adenoma and gene detection. The MEN1 patient’s case was complicated with relapsed primary hyperparathyroidism (PHPT), and they received ultrasound-guided radiofrequency ablation (RFA). The patient had a remarkable recovery after RFA treatment for the relapsed PHPT. It might be an alternative treatment for MEN1 patients with poor conditions such as high surgical risk, unwillingness to choose parathyroid surgery or those unable to tolerate surgery. Individualized therapy significantly benefits the prognosis of MEN1 patients.

##  


Figure 1Low-power view of pancreatic neuroendocrine tumor by immunostaining. Neuroendocrine tumor (G1), lymph node metastasis: around common hepatic artery (3+/5), around pancreas head (1+/3). Immunohistochemistry: (lymph nodes E): CgA (+), Syn (+), NSE (+), CD56 (+), CKpan (+), Ki-67 approximately 2%; (lymph nodes Q): CKpan (+), CD 56 (+), CgA (+), NSE (+), Syn (+), Ki-67 < 1%, Glucagon (+), insulin (−), somatostatin (−). CgA: Chromogranin A; Syn: Synaptophysin; NSE: Neuron-specific enolase. Multiple endocrine neoplasia type 1 (MEN1) is an autosomal dominant genetic disease with a prevalence of about 2~3/100,000 [1], which is typically characterized by a higher occurrence of pituitary neoplasia, primary hyperparathyroidism and gastroenteropancreatic neuroendocrine tumors [2,3]. Currently, the pathogenesis of MEN1 is considered to be mutations in the MEN1 gene [4]. Here, we report a MEN1 patient, who suffered recurrent primary hyperparathyroidism (PHPT) after the initial surgery, and had his recurrent PHPT treated with ultrasound-guided radiofrequency ablation (RFA). The patient was a 52-year-old Han Chinese male, whose grandmother was intermarried to her cousin. His brother and cousin died of thymic carcinoid with bone metastasis and neuroendocrine neoplasms with liver metastasis, respectively. Seven years ago, due to recurrent upper abdominal pain, peptic ulcer and urinary calculi, the patient was found in the following physical condition: a level of gastrin 768.73 pg/mL (reference ranges, 13–115 pg/mL), albumin-corrected serum calcium 2.85 mmol/L (reference ranges, 2.11–2.52 mmol/L) and intact parathyroid hormone (iPTH) 163 pg/mL (reference ranges, 11.0–67.0 pg/mL); both parathyroid ultrasonography and emission computed tomography (ECT) indicated a lesion of right inferior parathyroid. The patient was diagnosed with PHPT and underwent endoscopic-assisted right inferior parathyroidectomy in August 2014. The patient had not been diagnosed with MEN1 at the time of the single gland resection of parathyroid gland, and only single gland lesion as the initial treatment of PHPT. The pathological sections showed right inferior parathyroid adenoma. Although the serum calcium returned to normal after surgery, the peptic ulcer was still not cured, and the gastrin level was significantly higher than the normal value. The subsequent genetic testing pointed to mutations of the MEN1 gene, CDS5 sub-region, c.866C>A nucleotide variation and p.Ala289Glu amino acid variation (Table 1). The patient underwent laparoscopic pancreaticoduodenectomy with total meso-pancreas excision in December 2014. The pathologic results showed neuroendocrine neoplasm in histological type (G1) (Figure 1). The patient had been diagnosed with gastrinoma according to clinical manifestation and immunohistochemistry results.
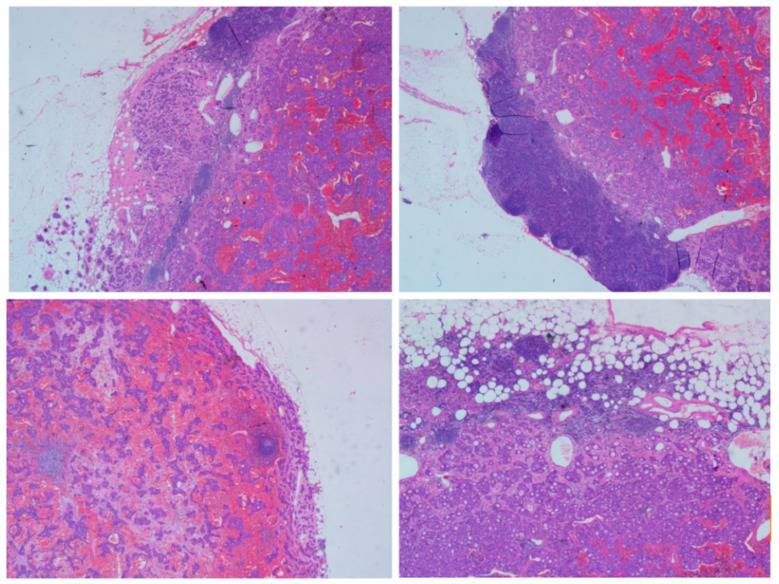

diagnostics-12-02553-t001_Table 1Table 1MEN1 genetic detecting analysis.GeneMEN1 NM_130803Inheritance modeAutosomal dominant (AD)Gene subregionCDS5Nucleotide alterationc.866C>AAmino acid alterationp.Ala289GluFunctional changeMissenseHomozygous/heterozygousHeterozygous
Figure 2The image of the parathyroid glands in 99mTc MIBI scintigraphy; MIBI accumulation was observed at 120 min as well as at 30 min in neck lesion which was thought to be parathyroid lesion (arrows in both panels). Gastrin levels returned to normal after surgery, and the patient took Esomeprazole 20 mg/d for 7 years, with regular follow-up. The latest reexamination showed a gradual increase in his serum calcium. The latest albumin-corrected serum calcium was 2.63 mmol/L (reference ranges, 2.11–2.52 mmol/L), serum phosphorus level was 0.72 mmol/L (reference ranges, 0.85–1.51 mmol/L) and PTH level was 141 pg/mL (reference ranges, 11.0–67.0 pg/mL). Parathyroid gland ECT indicated a slight increase of delayed uptake tracer in the right middle and upper pole of the thyroid, left middle and upper pole of the thyroid, and left lower pole of the thyroid, suggesting possible parathyroid origin (Figure 2). Dual-energy X-ray examination suggested osteoporosis (T value of left femoral neck −2.8).
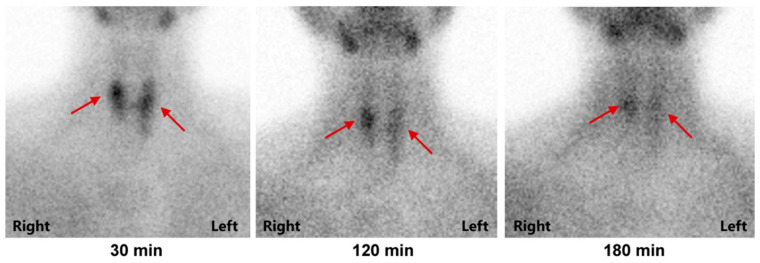

Figure 3Changes of two left parathyroid glands before and after RFA by neck ultrasonography. US examination revealed 6 mm × 3 mm × 7 mm (**A**) and 9 mm × 6 mm × 3 mm (**C**) parathyroid nodules posterior to the left lobe of the thyroid gland pre-RFA. Two months after RFA, nonenhancement area (arrows) covered PHPT nodules on contrast-enhanced ultrasound (CEUS) (**B**,**D**), and the volume of the nodules decreased to about 0.01 mL (3.6 mm × 2.5 mm × 2.8 mm, superior left) (**B**) and 0.02 mL (4.6 mm × 2.8 mm × 2.7 mm, inferior left) (**D**). Ultimately, the patient was diagnosed with MEN1, based on their medical history and examination data as follows: (1) gastrinoma; (2) multiple parathyroid adenoma; (3) c.866C>A nucleotide variation in MEN1 gene. As the patient had multiple parathyroid lesions, surgery was one of main choices, including subtotal parathyroidectomy or total parathyroidectomy and parathyroid forearm transplantation. The patient was fully informed before treatment that the preferred treatment option was surgical resection. However, the patient had strong resistance to surgery since he had undergone multiple surgeries. A prospective study showed that complete ablation was achieved in 38 of the 39 nodules in the 39 enrolled participants in RFA in patients with PHPT. Recurrent laryngeal nerve paralysis occurred in 5.1% of the patients, who recovered spontaneously within three months [5]. These data indicate that RFA might be an alternative therapy for patients who cannot tolerate undergoing surgery. Due to postoperative recurrence, the patient underwent RFA of two left parathyroids under the ultrasound guidance and saw remarkably reduced dimensions of parathyroids (Figure 3). The patient had no hoarseness, no local pressure and no neck pain after RFA. In the latest follow-up, the patient’s serum calcium had declined to within the normal range (2.45 mmol/L at Day 60 follow-up after RFA), and the parathyroid hormone (PTH) concentration declined from 141.0 pg/mL to 81.8 pg/mL (Table 2). More prolonged follow-up is necessary. In this report, we showed a case of MEN 1 with a germline frameshift mutation (c.866C>A in MEN1 gene) accompanied by multiple parathyroid adenomas and gastrinoma. A diagnosis of MEN1 was established in an individual by one of three criteria [2]. Firstly, on the basis of the occurrence of two or more primary MEN1-associated endocrine tumors, such as parathyroid adenoma, enteropancreatic tumor and pituitary adenoma. Secondly, the occurrence of the MEN1-associated tumors in a first-degree relative of a patient with a clinical diagnosis of MEN1. Thirdly, the identification of a germline MEN1 mutation in an individual, who might be asymptomatic and has not yet developed serum biochemical or radiological abnormalities indicative of tumor development. The patient met the two diagnostic criteria and was diagnosed with MEN1. Systemic screening of endocrine organ functions and imaging was performed, and no other lesions except parathyroid adenoma and gastrinoma were found. After the biochemical diagnosis of PHPT in MEN1 patients, surgical indications are similar to those of sporadic PHPT, including evidence of symptomatic or significant hypercalcemia, renal calculi and bone diseases such as decreased bone mineral density or fracture [2]. The surgical methods of MEN1 with PHPT remain controversial. Several studies showed that nearly 42% patients with less than three parathyroidectomies still suffer from PHPT after the surgical treatment, while only 12% of patients with more than three parathyroidectomies have persistent PHPT. Nevertheless, they have a higher risk of permanent hypothyroidism [6,7,8,9]. Furthermore, patients with the MEN1 gene mutation are prone to relapse of PHPT. After the parathyroidectomy, scar tissue around the parathyroid gland makes the second surgery more difficult for patients with recurrence, and the complication rate during reoperation is higher than during the primary surgery [10]. The patient’s history of parathyroid surgery seven years ago introduced even more difficulties to our treatment in this instance. In the management of PHPT, thermal ablation has been recommended as an alternative in the last ten years [11,12]. Thermal ablation resulted in less estimated blood loss, shorter treatment time, no scars on the neck and required only local anesthesia compared to surgery [13]. It was found that the cure rate of thermal ablation could reach nearly 95%, which is comparable to surgery in PHPT patients [12]. Compared with the incidence of permanent recurrent laryngeal nerve injury in parathyroidectomy, the rate of permanent nerve palsy hoarseness was lower in thermal ablation (0.8% vs. 3.9%) [13,14]. The patient chose ultrasound-guided RFA to reduce the trauma and risk of reoperation, since many clinical studies showed the efficacy and safety of thermal ablation for PHPT, coupled with our successful experience in treating PHPT with RFA. The serum calcium was reduced to the normal range merely two days after RFA with satisfactory efficacy and safety. Individualized therapy significantly benefits the prognosis of MEN1 patients.
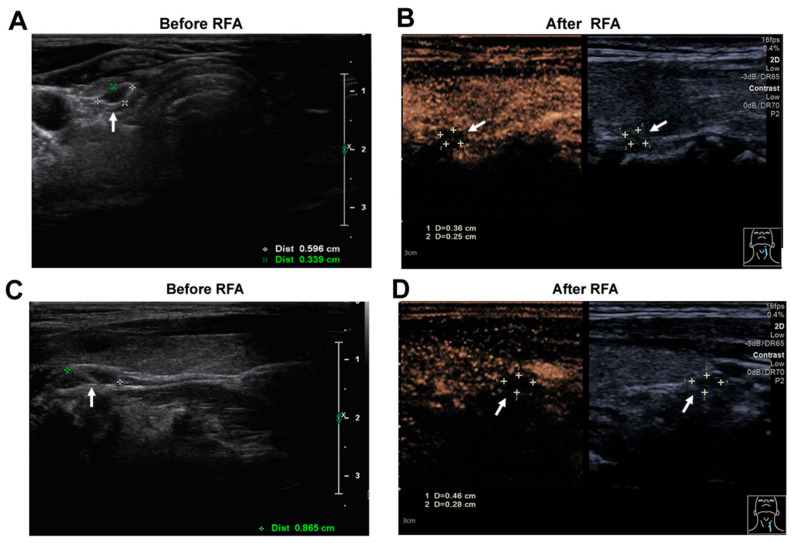

diagnostics-12-02553-t002_Table 2Table 2The patient’s serum calcium and PTH changes before and after RFA.Administration Time (Day)Serum Calcium (mmol/L)PTH (pg/mL)Before RFA2.63141.0Day 0 after RFA2.7132.8Day 1 after RFA2.5260.8Day 2 after RFA2.481.6Day 60 after RFA2.4581.8


## Data Availability

Not applicable.

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
