# Peer review of "Ultrasound-Guided Radiofrequency Ablation for Primary Hyperparathyroidism Induced by Multiple Endocrine Neoplasia 1—A Case Report"

_diagnostics, 2022, doi:10.3390/diagnostics12102553_

Round 1

Reviewer 1 Report

This case report proposes an interesting alternative method to manage recurrence of primary hyperparathyroidism (PHPT) in MEN1 patients. Indeed, the management of PHPT in these patients is debated.

Nevertheless, there are some points to be clarified.

MEN1 is characterized by neuroendocrine neoplasms (not generic pancreas tumor as stated), thus I suggest being more specific and citing the following recent paper by Ruggeri et al. (10.1007/s40618-022-01905-4).

Diagnostic criteria of MEN1 should be discussed, as even before genetic testing, the familial history and clinical manifestations suggested the diagnosis.

Please provide normal range value of each parameter (gastrin, calcium, PTH…) since the beginning of case presentation. Were ionized or albumin-corrected calcium levels available?

The choice to perform single gland excision as initial management of PHPT should be detailed, as other surgical approaches have been proposed in guidelines. Similarly, the choice to perform RFA after recurrence should be explained, as it seems that it was just a patient preference. Furthermore, the length of follow up after surgery should be specified. In particular, it seems that already 60 days after RFA (Tab 2), PTH levels are above the normal range.

In figure 2, there are no arrows unlike stated in the caption.

Reviewer 2 Report

In the abstract and on page 3 you talk about a gastrinoma. However, the operation in 2014 was a panreaticoduodenoectomy resulting in a neuroendocrine tumor. Where does the gastrinoma come from? The figure 1 describes a NET G1 in my opinion it lacks the indication of the KI-67 percentage is missing.

There are small typing/layout errors, e. g. on page 1, in the text line 10 distance at “metastasis and”. There is also a typo in the figure 2 descrition, it should be "MIBI" instead of "MIBC".

The image description of Figure 2 indicates arrows, which do not appear on the image. In addition, one sees in the picture a tracer retention on the right cranial and not just one on the left thyroid lobe. Are there no SPECT pictures?

In the discussion (last paragraph) I miss a statement on the problem of size and imaging parathyroids in sonography. The RFA of such a small lesion is possible but not harmless and therefore I am rather skeptical about the snumbers given for recurrence paresis (quotations 11 and 12).
Also the limitation of the measurement should be discussed, since a volume reduction of 0.01ml is a good guess for me rather than proven by measurement!

Round 2

Reviewer 1 Report

The authors have addressed all the comments and suggestions.